# Tinea Capitis Induced by Barber Shaving: Isolation of *Trichophyton tonsurans*

**DOI:** 10.3390/jcm14020622

**Published:** 2025-01-19

**Authors:** Giampaolo Addari, Marialuisa Corbeddu, Cristina Mugheddu, Mariangela Chessa, Grazia Vivanet, Caterina Ferreli, Laura Atzori

**Affiliations:** 1UOC of Dermatology, AOU Cagliari, via Ospedale 54, 09126 Cagliari, Italy; promedca.giampaolo@gmail.com (G.A.); m.corbeddu@aoucagliari.it (M.C.); c.mugheddu@aoucagliari.it (C.M.); ma.chessa@aoucagliari.it (M.C.); graziavivanet@gmail.com (G.V.); ferreli@unica.it (C.F.); 2School of Specialization of Dermatology and Venereology, Department Medical Sciences and Public Health, University of Cagliari, 09124 Cagliari, Italy

**Keywords:** *Trichophyton tonsurans*, barber, shaving, *tinea capitis*, hairdressing, scarring alopecia, *kerion celsi*

## Abstract

**Background/Objective:** *Tinea capitis* is a common scalp fungal infection with significant implications for public health, particularly in regions where proper hygiene and access to healthcare are limited. Emerging evidence suggests that this disease, particularly in young male individuals, may be related to certain unsanitary practices in barbershop settings, such as the use of contaminated shaving equipment. To increase awareness of the risk of scalp dermatophyte infections by disclosing different cases of *tinea capitis* that had arisen shortly after hairdressing procedures and providing a comprehensive review of the existing literature. **Patients and Methods**: 10 cases of young, adult male patients developed tinea capitis after haircuts carried out at different local barbershops in Sardinia. A collection of data regarding age, sex, location of the disease, clinical manifestations as well as direct microscopy and cultural investigations were performed. **Results:** Clinical manifestations varied among patients, exhibiting both non-inflammatory and inflammatory features, cultural investigations were positive for *Trichophyton tonsurans*. Patients were treated with griseofulvin or terbinafine in combination with topical antimycotics. Two cases out of the ten patients developed scarring alopecia. **Conclusions:** Outbreaks of *T. tonsurans*-induced *tinea capitis*, linked to hairdressing, have been recorded in many different countries. Timely diagnosis and therapy are crucial, since any delay can result in disease dissemination and potential complications such as scarring alopecia, particularly in the inflammatory forms.

## 1. Introduction

Dermatophytosis is a worldwide fungal infection that affects 20–25% of the general population, most commonly in children between 3 and 7 years, likely due to differences in sebaceous secretion and skin pH compared to the pubertal age and adulthood [1,2].

*Tinea capitis* is a form of dermatophytosis, prevalent in rural and suburban areas, accounting for 4–10% of all such infections. It affects both sexes and is more commonly associated with low socioeconomic status [1,2]. It is a cause of public health concern due to its highly contagious nature, social stigma and potential complications such as *kerion celsi* and scarring alopecia [3].

The etiologic agents responsible for tinea capitis include *Microsporum* and *Trichophyton* species. The prevalence of these diseases vary across geographical regions, ethnic groups, and climates [4]. While the zoophilic fungus, *Microsporum canis*, remains the predominant causative agent worldwide; over the past century, epidemiological research has observed the emergence of *Trichophyton tonsurans* as the predominant causative agent of tinea capitis in many different countries, including the United States and other European nations [2,5,6,7].

The reasons underlying the increasing prevalence of *T. tonsurans*-induced *tinea capitis* include migration patterns from West Africa, Latin America, or the Caribbean, where the disease appears to be more prevalent. Additionally, the widespread use of the antifungal griseofulvin, which is more suitable for treating *Microsporum* rather than *Trichophyton tinea capitis* infections, may have played a role in selecting this fungus, favoring its spread due to inadequate therapy [3,5,8]. Furthermore, the recent increasing resistance of *Trichophyton* genus to terbinafine may be considered another alarming factor responsible for its increased dissemination [9].

Interestingly, recent reports have highlighted an upward-running incidence of *Trichophyton tonsurans* infections in young males and adults who frequently visit barbershops for haircuts and hairdressing, suggesting a potential role of these settings in the spread of this pathogen [5].

This paper reports additional cases of *Trichophyton tonsurans*-induced tinea capitis in Sardinia, providing similar instances described in the literature and analyzing the potential underlying factors that contribute to its transmission.

## 2. Patients and Methods

Our study identified 10 young, adult male patients who presented to our dermatologic outpatient clinic for the onset of single and multiple, inflammatory and non-inflammatory lesions primarily affecting the scalp, nape, or cervical region.

The first patient was a 25-year-old male who reported the development of a single erythematous, severely inflamed plaque with serosanguineous and occasional purulent discharge on the right temporal region, accompanied by fever and painful lymphadenopathy. General malaise and discomfort were significantly impacting his quality of life, prompting him to seek medical attention.

The severe clinical presentation initially suggested the diagnosis of a bacterial sycosis of the scalp or impetigo. Accordingly, the patient was empirically treated with oral antibiotics for two weeks, without showing any significant improvement.

In the following weeks, two acquaintances of the first patient presented to our clinic with analogous complaints, manifesting in this case pruritic, well-circumscribed, erythematous-scaly patches on the scalp and cervical region.

Clinical and dermoscopic features were this time more suggestive of a tinea capitis infection; therefore, a direct microscopic examination was performed in the affected areas, confirming the presence of fungal spores and hyphae.

At this point, a detailed medical history was collected from all the three patients and revealed that the only common factor among them was their recent visit to the same local barbershop, where they had their hair cut on a regular basis.

Consequently, the patients were treated with oral antimycotic drugs, showing a complete resolution of their symptoms within 4–6 weeks, with the only exception being the first patient who, due to the initial severe manifestations, developed a scarring alopecia.

Over the following weeks, additional patients were evaluated in our clinic, showing a similar onset of scalp and neck lesions with a history of a barber trim or shave prior to the appearance of their symptoms. Some of them also mentioned seeing other patrons with similar presentations after their visit to the same barbershop, although in different facilities from the previous cases.

Therefore, we proceeded to systematically collect data related to the clinical characteristics, dermoscopic findings, microscopic analyses and fungal cultures of the skin and hair specimens of the affected patients. This comprehensive approach aimed to isolate the causative agent responsible for these cases and break the chain of infections by informing barbers about the potential contamination of their shaving equipment.

All of the 10 patients included in our study underwent a routine Wood’s lamp examination (negative in all cases) and a dermoscopic examination. Microscopic evaluation and fungal cultures were conducted in 8/10 and 3/10 patients, respectively. The inclusion criterion was a clear history of a recent barber shave prior to the onset of clinical manifestations.

## 3. Results

Patients’ cohort profiles with clinical, microscopic, and cultural results are summarized in Table 1.

### 3.1. Patients’ Cohort Profile

The patients’ cohort had a median age of 23.5 years, with the youngest individual being 17 years old and the oldest being 68 years old.

All patients were male, reflecting a potential association with barber-related buzz cuts in this gender.

The median duration between the suspected barbershop exposure and the onset of the clinical manifestations was 7 days, while the median time of diagnosis after the onset of the symptoms was 22 days.

The main affected location was the occipital region, followed by the nape. Other cases developed lesions on the temporal region, neck, beard and on the upper back.

### 3.2. Clinical Manifestations

Among these patients, different clinical manifestations were noted, ranging from typical tinea capitis presentations with erythematous-scaling patches to more severe forms of *Kerion Celsi,* characterized by large, oedematous, and elevated plaques with occasional pus discharge.

The following clinical presentations were observed across the patients’ cohort:Two out of the ten individuals in the study exhibited single or multiple erythematous, inflamed, and painful plaques, with occasional serosanguineous or suppurative discharge, associated with fever and lymphadenopathy. Intense scalp discomfort and distress were deeply impacting their quality of life (Figure 1).Eight out of the ten patients manifested single or multiple, round, scaly and erythematous patches of the scalp, mainly located in the occipital and nape regions. Since these latter were the most affected sites, delays in seeking medical attention were typical, with the subsequent progression of the primary lesions (Figure 2). Mild to moderate pruritus was the referred common symptom in these cases; however, one patient exhibited a completely asymptomatic presentation (Figure 3).

### 3.3. Dermoscopic, Microscopic and Cultural Findings

Dermoscopy revealed in most of the cases an erythematous background with scales and different aspects of broken hairs, comma hairs, or corkscrew hairs within the patches, more indicative of a *Trichophyton* infection [10] (Figure 4).

In the inflammatory forms, significant erythema, yellow crusts, and pustules were also noted.

Microscopic examination of the removed hair shafts and scraped scalp scales was performed in 8/10 patients using standard preparation techniques with KOH 20% [4].

The collected samples from the scalp and nape lesions were placed on clean glass slides and immersed in three drops of 20% potassium hydroxide solution. After applying a cover slip, the samples were examined under a microscope 15 to 30 min later to interpret the findings. The samples disclosed the presence of fungal hyphae and spores within the hair stubs, exhibiting an endotrix pattern more indicative of *Trichophyton* infections [11] (Figure 5).

Finally, fungal cultures from scalp specimens obtained through moistened sterile cotton swabs and grown on Sabouraud dextrose agar media confirmed the presence of the dermatophyte *Trichophyton tonsurans* in all the three cases performed in this study (Figure 6).

### 3.4. Treatment and Follow-Up

All the patients were treated with oral antifungal therapy, typically terbinafine at a dosage of 250 mg/daily for 4–6 weeks, or griseofulvin 500 mg/daily for 4 weeks, in combination with topical antifungal treatment. Although our laboratory was unable to perform specific antifungal susceptibility testing, the clinical response observed in our cases, with complete resolution of the symptoms within 4–6 weeks, seems to currently exclude antifungal resistance as a major issue in our current clinical setting. Topical antifungals were also prescribed to rapidly reduce the fungal load, and the risk of spread, as well as to diminish the risk of re-infection. They included lotions and scalp cleansers based on ciclopiroxolamine to provide a different active principle from the oral medication, and for the wider spectrum of this molecule, which covers some bacteria and yeasts.

Nevertheless, the two patients who exhibited the more pronounced inflammatory presentations developed permanent scarring alopecia as a final outcome (Figure 7).

Fortunately, no other family members or close contacts of the index patients were reported to have developed similar dermatological conditions, which could have been a potential issue due to the anthropophilic nature of *T. tonsurans*.

## 4. Discussion

This study highlights a series of tinea capitis infections in the context of barbershop exposure, where *Trichophyton tonsurans* was identified as the responsible agent. Over the past century, this anthropophilic dermatophyte has become the predominant causative agent of *tinea capitis* in several countries, such as the USA, England, Belgium, and the Netherlands, mainly replacing the previously more common *Microsporum* infection [7,12,13,14].

While *Microsporum* remains the prevalent fungal agent worldwide for tinea capitis, an increasing trend of this infection caused by anthropophilic fungi, including *T. violaceum*, *T. soudanense* and especially *T. tonsurans*, has been observed and is expected to continue.

Although the exact causes of the upward-running incidence of *T. tonsurans*-induced tinea capitis remain not fully understood, several hypotheses have been proposed, including:-The association with migration patterns from West Africa, Latin America, or the Caribbean, where the disease appears to be more prevalent, or with wrestling sport competitions in the United States [5,6,7].-The widespread use of over-the-counter antifungal products containing a triple combination of antibacterial, steroid, and antifungal agents; the extensive administration of griseofulvin, which is not the first choice for *Trichophyton*-induced tinea capitis infections, coupled with the emergence of drug-resistant strains due to mutations in the squalene epoxidase gene, may be additional driving factors of the increasing resistance and prevalence of *Trichophyton* species [5,9,15].-Environmental persistence: Although *T. tonsurans* is an anthropophilic fungus that primarily infects humans, several studies have detected its presence on different fomites, such as theater seats, hairbrushes, combs, bedding, and caps, suggesting the possibility of an indirect transmission in addition to the direct person to person contact [14,16]. Additionally, the fungus can remain viable in the environment for extended periods under certain conditions, which may explain the persistence and recurrence of the infection within families, as noted in previous studies [17].-The different clinical presentations associated with *T. tonsurans* infections can further contribute to its widespread dissemination, as they may delay timely and accurate diagnosis, promoting the spread of the disease and rendering eradication more challenging. Clinical signs can be extremely variable and can mimic other dermatological conditions such as impetigo, seborrheic dermatitis, psoriasis, hair tuft folliculitis, or even dissecting cellulitis [14,18,19].-Moreover, individuals with a limited number of infected hairs may act as asymptomatic carriers, enabling the dissemination of the disease to other vulnerable populations [20]. Consequently, in every *T. tonsurans*-induced tinea capitis case, a thorough examination of the scalp is crucial to rule out the presence of a disease reservoir. Carrying out investigations of the entire family, as well as inspecting any shared equipment, is strongly recommended to identify and eliminate any potential source of the infection [20].

Following these premises, barbershop settings may represent an ideal environment for the spread of these fungal infections. The use of contaminated shaving equipment, the traumatic nature of haircut procedures, the shared use of barber chairs, and the lack of adherence to proper hygiene in these establishments may play a particular role in the transmission of these diseases.

Among these factors, contaminated barbershop tools are particularly concerning, as they may act as transmissive carriers for fungal spores, particularly in children and young adults, who frequently undergo haircuts and grooming procedures to emulate popular hairstyles [21].

The clinical manifestations observed in our cohort of patients, along with study variables, such as age, sex, disease location, and the isolated fungal agent, are in accordance with those recently described in the literature, as reported in Table 2.

The display of different clinical manifestations, ranging from mild forms of *tinea capitis* to more severe inflammatory presentations, may suggest different mechanisms of host immune response to dermatophyte infection as well as a potential role of local trauma induced by the shaving process [5]. The stereotypical involvement of the occipital, nape, and temporal regions suggests that repetitive shaving actions, commonly performed in these sites, may facilitate prolonged contact between fungal spores and the scalp, potentially driving them deeper into the skin and triggering more inflammatory lesions if the shave is particularly traumatic. Additionally, the premature application of corticosteroids without prior identification of the underlying disease may worsen the clinical course, potentially leading to more severe inflammatory reactions. Nevertheless, in some severe forms of *Kerion Celsi*, a short course of oral corticosteroids, such as prednisone, may be administered to lower the inflammatory burden. In our cases, the improvement with the antifungal therapy alone was rapid, not requiring additional treatment, but the scarring damage had already occurred.

Based on the data gathered from other healthcare facilities and patient medical histories, our cases likely represent just a fraction of the total number of individuals affected by barber-related *tinea capitis* infections. Therefore, it is crucial to be aware of this condition and to include it in the differential diagnosis, as timely recognition can prevent outbreaks and complications, such as scarring alopecia.

The limitations of our study are related to the small sample size, as most patients were referred to other outpatient clinics, hindering comprehensive data collection. Additionally, the performance of the culture test in only three out of the ten studied cases is another limiting factor, as it does not guarantee that *T. tonsurans* is the only etiological agent involved in this study. In the future, larger and more in-depth studies with an adequate number of fungal isolations are needed to establish the real impact and prevalence of this dermatophyte in our region.

## 5. Conclusions

*Tinea capitis* induced by barber shaving is a condition that warrants greater awareness, as it may represent a significant public health concern. The high contagiousness of the disease, exacerbated by the trauma of shaving, and the potential presence of asymptomatic carriers can easily spread the infection, increasing the risk of local outbreaks. To effectively prevent the transmission of these infections, it is essential to instruct and educate barbers regarding their responsibility in maintaining high hygiene standards, employing, whenever possible, single-use equipment or by sterilizing shaving tools between clients. In case of suspected infection, patients should be advised not to share any personal items such as combs, towels, brushes, or pillows with close contacts, as well as to inform their local health department for investigation and the prevention of further cases.

## Figures and Tables

**Figure 1 jcm-14-00622-f001:**
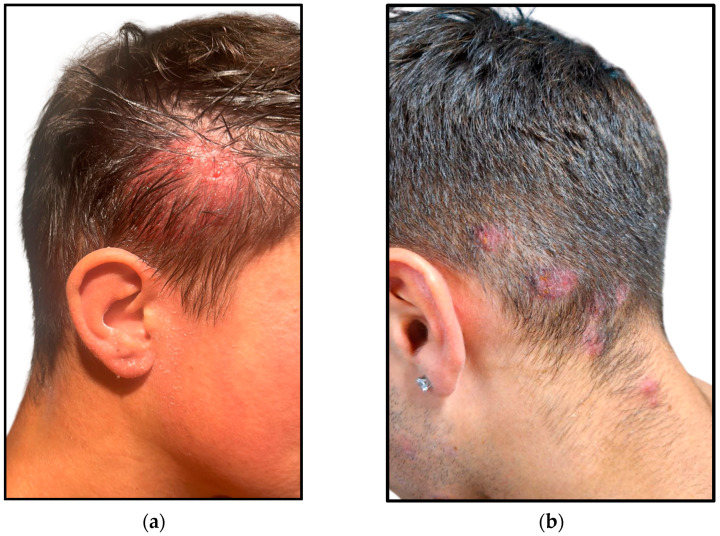
Inflamed lesions of the scalp: (**a**) Single, erythematous, boggy, and elevated plaque on the temporal region with serosanguineous discharge; (**b**) Multiple erythematous, elevated, coin-shaped lesions on the occipital and nape regions.

**Figure 2 jcm-14-00622-f002:**
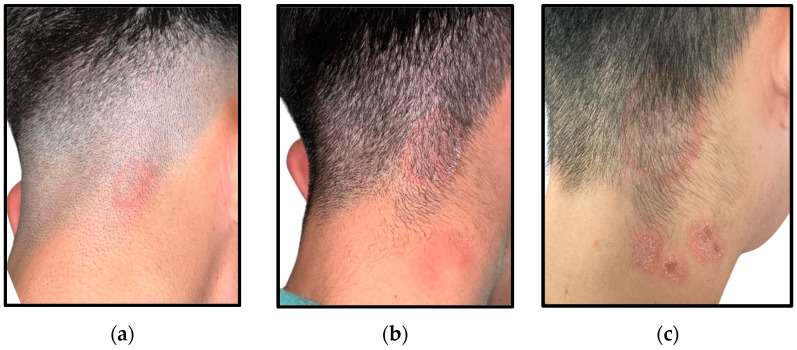
Development and progression of an erythematous, ring-like lesion appeared on the nape area of a young patient after a recent haircut at the local barber: (**a**) Single patch on the nape region with peripheral red halo ring and central resolution; (**b**) Progressive enlargement of the original patch, with concomitant onset of three additional reddish lesions on the posterior cervical region; (**c**) Final aspect of the four patches right before the diagnosis.

**Figure 3 jcm-14-00622-f003:**
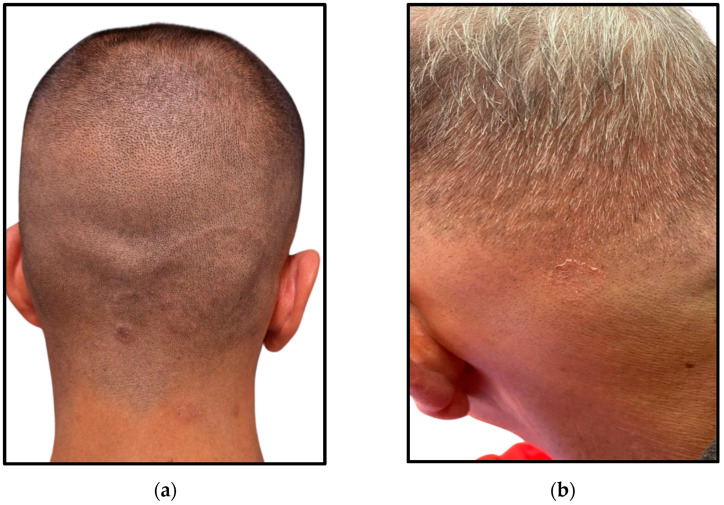
Other clinical aspects: (**a**) Round, enlarged patches with peripheral, elevated border on the occipital region of a young patient; (**b**) Small, single, scaly, and erythematous patch on the occipital region of a 68-year-old patient who visited the same barbershop as the other younger patrons. This was the only asymptomatic case.

**Figure 4 jcm-14-00622-f004:**
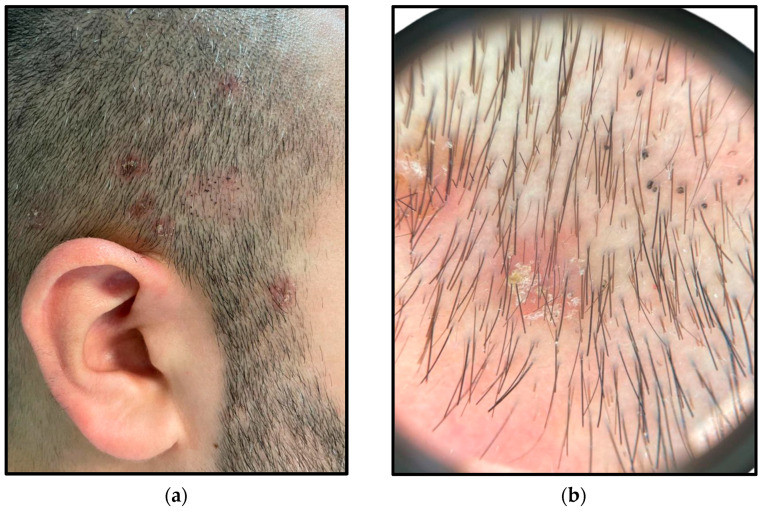
Erythematous and alopecic patches on the temporal region: (**a**) Of note, the clinical appearance of an alopecic patch with black dots; (**b**) Dermoscopy of the lesions showing different aspects of broken hairs, comma hairs, and corkscrew hairs; erythema and faint scales are also present.

**Figure 5 jcm-14-00622-f005:**
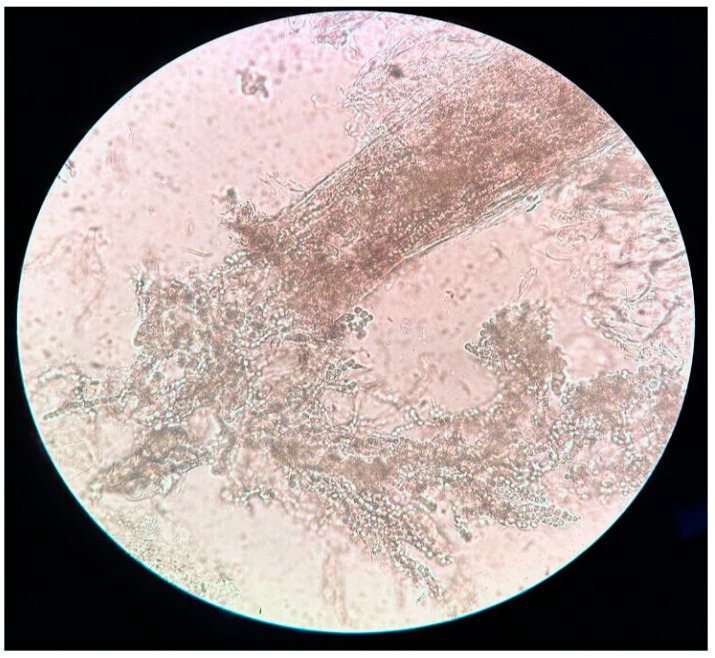
Representation of intrapilar arthroconidia filling the hair shaft, indicative of a *Tricophyton* infection. The frayed tip is the result of the fungal infection, which makes the hair brittle and prone to rupture.

**Figure 6 jcm-14-00622-f006:**
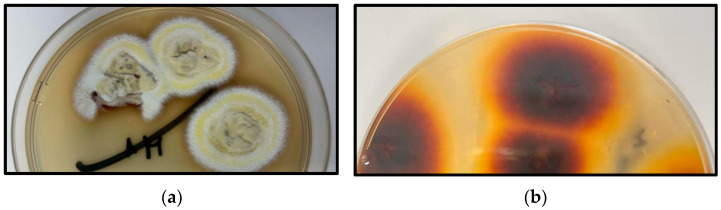
Cultures of *Trichophyton tonsurans* on Sabouraud agar: (**a**) Top side; (**b**) bottom side.

**Figure 7 jcm-14-00622-f007:**
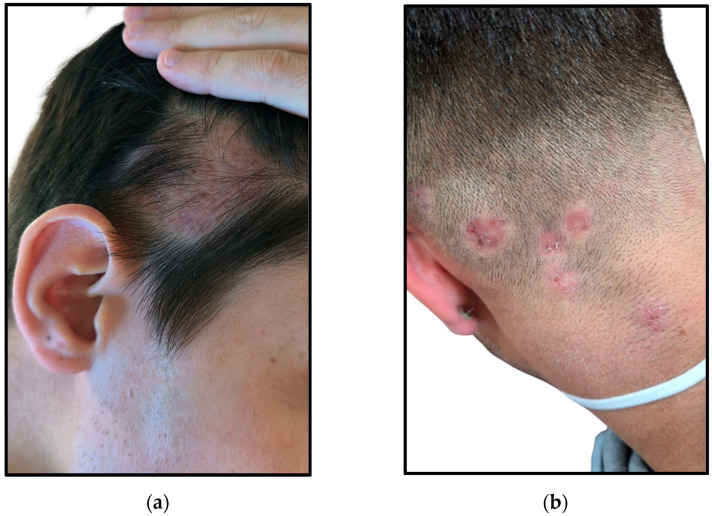
Outcome with permanent scarring alopecia of the two patients depicted in Figure 1: (**a**) Single alopecic area on the temporal region; (**b**) Multiple alopecic areas on the occipital and nape regions.

**Table 1 jcm-14-00622-t001:** Patients’ cohort profiles with clinical, microscopic, and cultural results.

**Demographics**	**Patient 1**	**Patient 2**	**Patient 3**	**Patient 4**	**Patient 5**
Age (MD ^1^: 23.5 years)	25 years	20 years	22 years	24 years	26 years
Sex	M	M	M	M	M
Location of the disease	Temporal region	Nape, neck	Nape, neck	Nape, beard, upper back	Nape, neck
Clinical manifestations	Kerion Celsi and idic reaction	Erythematous-scaly patches with central resolution	Erythematous-scaly patches with central resolution	Erythematous-scaly patches with central resolution	Erythematous-scaly patches
Symptoms	Pain, fever, lymphadenopathy	Pruritus	Pruritus	Pruritus	Pruritus
Onset after shaving (MD: 7 days)	7 days	2 days	28 days	14 days	7 days
Time of diagnosis after onset (MD: 22 days)	30 days	14 days	45 days	60 days	10 days
Microscopy	Not performed	Positive	Positive	Positive	Positive
Culture	Not performed	Not performed	Not performed	Positive	Positive
Therapy	Oral antibiotic, then oral griseofulvin	Oral terbinafine and topical antifungal	Oral terbinafine and topical antifungal	Oral terbinafine and topical antifungal	Oral terbinafine and topical antifungal
Scarring Alopecia	Yes	No	No	No	No
**Demographics**	**Patient 6**	**Patient 7**	**Patient 8**	**Patient 9**	**Patient 10**
Age (MD ^1^: 23.5 years)	17 years	18 years	19 years	68 years	23 years
Sex	M	M	M	M	M
Location of the disease	Nape, neck	Temporal region,	Occipital region, nape	Occipital region	Nape, neck
Clinical manifestations	Erythematous-scaly patches	Alopecic patches with black dots	Kerion Celsi	Single erythematous-scaly patch	Single erythematous-scaly patch
Symptoms	Pruritus	Pruritus	Pain, fever, lymphadenopathy	Asymptomatic	Pruritus
Onset after shaving (MD: 7 days)	2 days	6 days	7 days	7 days	10 days
Time of diagnosis after onset (MD: 22 days)	7 days	50 days	14 days	7 days	30 days
Microscopy	+	+	+	-	+
Culture	+	Not performed	Not performed	Not performed	Not performed
Therapy	Oral terbinafine and topical antifungal	Oral griseofulvin	Oral terbinafine and topical antifungal	Antimycotic cleanser + topical antifungal	Oral terbinafine + topical antimycotic
Scarring Alopecia	No	No	Yes	No	No

^1^ MD: Median.

**Table 2 jcm-14-00622-t002:** Clinical characteristics and study variables of barber-induced tinea capitis in the literature.

	Russo et al. (2024) [22]	Galvañ-Pérez del Pulgar (2024) [23]	Bascón et al. (2023) [21]	Müller et al. (2020) [5]	Takwale et al. (2001) [24]
Number of patients in the study	29	29	107	18	2
Median age (years)	7.9 (3–14 y)	Young (age not specified)	19.7 (5–40 y)	11.5 (4–24 y)	71 y
Sex (percentage)	Male (79.3%) Female (20.7%)	Male > Female (unspecified ratio)	Male (99.1%) Female (0.9%)	Male (100%)	Female (100%)
Main disease location	Occipital and frontal regions of the scalp	Occipital and temporal regions of the scalp	Occipital and temporal regions of the scalp	Occipital region, neck, and beard	Scalp
Clinical manifestations	Pseudo- alopecic plaques with fine grayish flaking on the surface; Kerion celsi	Erythematous-scaly lesions; Inflammatory TC ^1^	Annular and scaly concentric patches; Kerion celsi	Annular plaques with central fading; inflammatory TC	Scaly scalp with hair loss
Most common isolated agent	*T. tonsurans*	*T. tonsurans*	*T. tonsurans*	*T. tonsurans*	*M. canis*
Cases of scarring alopecia (n° out of total patients)	None	Not specified	Not specified	1/18	None
History of barbershop visits before the onset of the disease (n° out of total patients)	Yes (28/29)	Yes (29/29)	Yes (107/107)	Yes (18/18)	Yes (2/2)

^1^ TC: Tinea Capitis; n°: Number.

## Data Availability

The data presented in this study are available on request from the corresponding author due to privacy restrictions.

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
