# Peer review of "Tinea Capitis Induced by Barber Shaving: Isolation of Trichophyton tonsurans"

_jcm, 2025, doi:10.3390/jcm14020622_

Round 1

Reviewer 1 Report

Comments and Suggestions for Authors

This is a well-written manuscript, however however, there are areas that could be improved. Below are my suggestions:

-this is a case series, rather than an original article; this should have been submitted as a "review", with a mini literature review in the Discussion part

-It would be useful to add the dosage of the antifungals administered (terbinafine and griseofulvin), as well as the duration of treatment

-Please explain the rationale behind administering topical antifungals, given that topical treatment is inefficient in tinea capitis (it cannot penetrate the hair shaft) - was it used adjuvantly? to reduce fungal load? to diminish the risk of re-infection? or to decrease the possibility of spread?

-Some papers advocate the adjunctive use of oral corticosteroids in cases of Kerion - this would be interesting to be added for the readers (line 277)

More dermoscopic pictures -if available- would be of great value to the readers, depicting other types of damaged hair, eg. Morse code hairs or clearly depicting corkscrew hairs 

Author Response

1) Reviewer: This is a case series, rather than an original article; this should have been submitted as a "review", with a mini literature review in the Discussion part.

1) Author's response: we thank the reviewer for the suggestion. We considered the proposal as a review of a clinical case series too ambitious, but if the chief editor accept the change of editorial format, we do agree with pleasure. Of course, we have provided the literature review in any case. 

In the discussion part, particularly in Table 2, line 266, we summurized the clinical characteristics, the main disease locations, the isolated fungi and the correlation with barbershops' exposure found in the literature. We opted for a table which allows, in a more concise fashion, to display the data we found.

2)Reviewer: It would be useful to add the dosage of the antifungals administered (terbinafine and griseofulvin), as well as the duration of treatment

2) Author's response: 

Thanks for pointing out the oversight, we have added a paragraph named 3.4 for the treatment details. 

3)Reviewer: Please explain the rationale behind administering topical antifungals, given that topical treatment is inefficient in tinea capitis (it cannot penetrate the hair shaft) - was it used adjuvantly? to reduce fungal load? to diminish the risk of re-infection? or to decrease the possibility of spread?

Authors: Italian guidelines for tinea capitis and widespread dermatophyte infection recommend topical treatment associeted with sytemics, and the reasons are almost all those indicated by the reviewers. We have added the following sentence in the manuscript, in the same paragraph 3.3 as above: "

All the patients were treated with oral antifungal therapy, typically terbinafine at the dosage of 250 mg/daily for 4-6 weeks or griseofulvin 500 mg daily for 4 weeks, in combination with topical antifungal treatment. Although our laboratory was unable to perform specific antifungal susceptibility testing, the clinical response observed in our cases, with complete resolution of the symptoms within 4-6 weeks, seems to currently exclude antifungal resistance as a major issue in our current clinical setting. Topical antifungals were also prescribed to rapidly reduce the fungal load, and the risk of spread, as well as to diminish the risk of re-infection. They included lotions and scalp cleansers based on ciclopiroxolamine to provide a different active principle from the oral medication, and for the wider spectrum of this molecule, that covers some bacteria and yeasts." 

4)Reviewer: Some papers advocate the adjunctive use of oral corticosteroids in cases of Kerion - this would be interesting to be added for the readers (line 277)

4) Author's response: This is a very controversial issue. While some authors advocate their use when inflammation is really severe; others, who had compared the oral corticosteroid + oral antifungal with the oral antifungal alone, did not find any statistic significant difference. We preferred not to add oral steroids in our cases, also because the improvement with antifungal was fast.  The final damage might be related to a certain delay in the access to care. 

We have added this sentence: "Nevertheless, in some severe forms of Kerion Celsi, a short course of oral corticosteroids, such as prednisone, may be administered to lower the inflammatory burden. In our cases, the improvement with the antifungal therapy alone was rapid, not requiring additional treatment, but the scarring damage had already occurred.

."

5) Reviewer: More dermoscopic pictures -if available- would be of great value to the readers, depicting other types of damaged hair, eg. Morse code hairs or clearly depicting corkscrew hairs

5)Author's response: we totally agree with this comment, unfortunately we have not additional good quality photos. Although dermoscopic examination is a routine procedure, taking pictures require a written consent that is not collected in the normal visit. The pictures provided for the article were taken once that the idea of performing the article had been concieved and a new case occurred.  Other photos we own are blurred or do not add any supplementary value to the article.

Reviewer 2 Report

Comments and Suggestions for Authors

Journal: JCM (ISSN 2077-0383)

Manuscript ID: jcm-3385914

Type: Article

Title: Tinea capitis induced by barber shaving: isolation of Trichophyton tonsurans.

Authors: Giampaolo Addari , Marialuisa Corbeddu , Cristina Mugheddu , Mariangela Chessa , Grazia Vivanet , Caterina Ferreli , Laura Atzori *

There is no need to number the keywords. I recommend writing the Latin words in Italic font also in the keyword section, e.g.  trichophyton tonsurans ->  Trichophyton tonsurans

The idea regarding the association between the increasing prevalence of T. tonsurans-induced tinea capitis and the widespread use of antifungal agents, such as griseofulvin, is not very clear in the paragraph below.

“The reasons underlying the increasing prevalence of T. tonsurans-induced tinea capitis include migration patterns from West Africa, Latin America, or the Caribbean, where the disease appears to be more prevalent, the widespread use of antifungal agents, such as griseofulvin, which is not considered the first choice for treating Trichophyton-related tinea capitis infections, as well as the increasing resistance of this species to terbinafine”.

In this paper, Addari and colleagues highlight a lesser-known issue in the context of hairdressing and barber salons, where the disinfection and sterilization of tools are of great importance. The research findings are surprising in this regard.

The paper is well illustrated with clinical images and lab results. Could you clarify if a routine examination with a Wood's lamp was performed?

I would also like clarifications regarding the oral and topical treatment, and whether blood tests were performed before and during the treatment.

I appreciate the originality of this paper in the context of an apparently ordinary topic with significant diagnostic and therapeutic implications.

Author Response

Thank you very much for taking the time to review this manuscript. Please find the detailed responses below and the corresponding corrections highlighted in the re-submitted files:

1) Reviewer: There is no need to number the keywords. I recommend writing the Latin words in Italic font also in the keyword section, e.g.  trichophyton tonsurans ->  Trichophyton tonsurans.

1) Author's response: Thank you for pointing it out. We have corrected trichophyton tonsurans with Trichophyton tonsurans, as suggested.

2) Reviewer: The idea regarding the association between the increasing prevalence of T. tonsurans-induced tinea capitis and the widespread use of antifungal agents, such as griseofulvin, is not very clear in the paragraph below.

“The reasons underlying the increasing prevalence of T. tonsurans-induced tinea capitis include migration patterns from West Africa, Latin America, or the Caribbean, where the disease appears to be more prevalent, the widespread use of antifungal agents, such as griseofulvin, which is not considered the first choice for treating Trichophyton-related tinea capitis infections, as well as the increasing resistance of this species to terbinafine”.

2) Author's response: thank you for the advice. We have modified the paragraph as follows:

"The reasons underlying the increasing prevalence of T. tonsurans-induced tinea capitis include migration patterns from West Africa, Latin America and the Caribbean, where the disease appears to be more prevalent; additionaly, the widespread use of the antifungal griseofulvin, which is more suitable for treating Microsporum rather than Trichophyton tinea capitis infections, may have played a role in selecting this fungus, favouring its spread due to inadequate therapy [3,5,8]. Furthermore, the recent increasing resistance of Trichophyton genus to terbinafine may be considered another alarming factor responsible of its increased dissemination [9]". 

3)Reviewer: The paper is well illustrated with clinical images and lab results. Could you clarify if a routine examination with a Wood's lamp was performed? 

3) Author's response: Thank you very much. Routine examination with Wood's lamp is routinely performed in our service, to immediately evaluate Microsporum canis cases. However, it was negative in all these patients, supporting the presence of Trychophyton species. A sentence was added in the text, among the performed assessment. 

4) Reviewer: I would also like clarifications regarding the oral and topical treatment, and whether blood tests were performed before and during the treatment.

4) Author's response: we did not perform blood test before or during treatment, as patients were very young and antifungal administration do not require any screening in Italy. Oral terbinafine was administered once daily (250 mg) for a variable period between 4 and 6 weeks, griseofulvin was administered 500 mg daily for 4 weeks. Topical antifungals included lotions and scalp cleansers based on ciclopiroxolamine to provide a different active principle, and for the wider spectrum of this molecule, that covers some bacteria and yeasts.

Round 2

Reviewer 1 Report

Comments and Suggestions for Authors

The authors have successfully addressed my comments. The paper has improved substantially and is suitable for publication in JCM.